

# Combining echocardiography with carotid ultrasound parameters in predicting major adverse cardiovascular events among older individuals with coronary artery disease

Tianli Jia[*], Xiaoxiao Li[*] and Qianqian Xie

Department of Ultrasound Medicine, Yantai Yuhuangding Hospital, Yantai, Shandong, China
[*] These authors contributed equally to this work.

## ABSTRACT

**Objective**. To investigate the predictive value of combining echocardiography with carotid ultrasound for major adverse cardiovascular events (MACE) among older individuals with coronary artery disease.

**Methods**. A total of 138 elderly individuals diagnosed with coronary artery disease and admitted to our facility from June 2020 to June 2021 were included in this study. These patients were categorized into two groups: a non-MACE group consisting of 84 patients and a MACE group with 54 patients, based on whether they experienced MACE within three years following their discharge. Additionally, a validation cohort of 141 patients was assembled, which was also divided into a non-MACE group with 90 patients and a MACE group with 51 patients. Upon admission, all participants underwent both transthoracic echocardiography and carotid ultrasound assessments. We then conducted a comparative analysis of the ultrasound parameters between the two groups.

**Results**. The MACE group had higher left ventricular end-diastolic diameter (LVEDD), carotid intima-media thickness (IMT), and Crouse score, and lower left ventricular ejection fraction (LVEF) and A-wave velocity (all $P < 0.05$). LVEDD, IMT, and Crouse score were risk factors for MACE, while LVEF and A value were protective factors against MACE. Derivation area under the curve (AUC) = 0.755; validation AUC = 0.754.

**Conclusion**. Echocardiography and carotid ultrasound parameters can be used to predict the occurrence of MACE within 3 years in elderly patients with coronary artery disease, and combined detection can effectively improve the accuracy of prediction.

## INTRODUCTION

Coronary artery disease is a prevalent cardiovascular condition among older adults, typically presenting with symptoms like discomfort or tightness in the chest, pain in

Corresponding author
Qianqian Xie, yaowen231@126.com

the chest area, and difficulties in breathing (*Alpert, 2023*; *Schwinger, 2023*; *Shaya et al., 2022*). The management of coronary heart disease (CHD) varies depending on the clinical scenario. For acute coronary syndromes (ACS), percutaneous coronary intervention (PCI) serves as the mainstay of treatment, while chronic coronary syndromes (CCS) are often managed medically. Post-PCI, patients are prone to myocardial ischemia-reperfusion, leading to major adverse cardiovascular events (MACE), which significantly affect the prognosis of coronary artery disease patients and increase their readmission rates and mortality (*Zeng et al., 2022*; *Figulla et al., 2020*). Consequently, it is imperative to discover proactive indicators that can forecast the severity and outcome of coronary artery disease. This proactive approach would facilitate early interventions aimed at diminishing the likelihood of experiencing MACE.

Ultrasound is a convenient, non-invasive, and highly repeatable imaging tool (*Picano et al., 2024*; *Truesdell et al., 2023*; *Upton et al., 2022*). Echocardiography can observe abnormal movements of the ventricular walls and assess the extent of these abnormalities, which may reflect the degree of myocardial ischemia and the extent of myocardial involvement in coronary artery disease (*Ximenes et al., 2023*). The carotid arteries are located superficially in the neck and serve as important conduits connecting the heart and brain. Carotid ultrasound is an essential parameter for evaluating the degree of arterial stenosis and the progression of atherosclerosis, and it has positive implications for guiding cardiovascular diseases (*Haberka et al., 2019*; *Mitchell et al., 2018*; *Song et al., 2021*). Currently, many studies both domestically and internationally use echocardiography and carotid ultrasound to evaluate cardiac function and carotid atherosclerotic lesions, and their diagnostic value has been proven (*Katsiki, Raggi & Korosoglou, 2022*; *Gallo, Mszar & Miname, 2022*). However, few studies have examined combined use for MACE prediction. Therefore, this study aims to investigate the predictive accuracy of combining echocardiography with carotid ultrasound examination parameters for predicting MACE in elderly patients with coronary artery disease, providing new insights into the prediction methods for MACE in coronary artery disease patients.

## METHODS

### Study subjects

This retrospective cohort study retrospectively analyzed coronary artery disease patients hospitalized for treatment between June 2020 and June 2021. Individuals who fulfilled the specified inclusion and exclusion criteria were chosen as participants for the study. Inclusion criteria: (1) Patients aged ≥60 years with chest pain symptoms within 24 h; (2) Patients diagnosed with acute myocardial infarction (AMI) or unstable angina (UA), both subtypes of ACS; (3) Patients with normal comprehension abilities willing to participate in follow-up after discharge. Exclusion criteria: (1) Individuals lacking complete clinical information; (2) Lost-to-follow-up patients (five cases); (3) Patients transferred to another hospital for treatment after diagnosis; (4) Patients with chest pain caused by other reasons such as trauma, generalized pain, rheumatic diseases; (5) Patients with comorbid immune, hematological diseases, hepatitis, pulmonary infections, psychiatric disorders, malignant

tumors; (6) Patients with a history of previous PCI or coronary artery bypass grafting (CABG); (7) Patients with pre-existing heart failure; (8) Patients with a history of previous stroke.

This study was approved by Yantai Yuhuangding Hospital's ethics committee. Given the retrospective nature of the study and the use of anonymized and de-identified data, the ethics committee waived the requirement for patient informed consent.

## Ultrasonic methods
### Echocardiography
Patients underwent ultrasound assessments while positioned on their left side and engaging in relaxed breathing. A color Doppler ultrasound device (m9cv, Mindray, China) operating at a probe frequency of 2.0–4.0 MHz was employed to systematically capture images in various views, including the parasternal short-axis, long-axis view of the left ventricle, two-chamber view, four-chamber apical view, and short-axis view of the left ventricle. To ensure standardization, all ultrasonographers received unified training on image acquisition protocols, and images were reviewed by a senior cardiologist for quality control. The data generated by the software analysis facilitated the calculation of several metrics, including left ventricular end-systolic diameter (LVESD), left ventricular end-diastolic diameter (LVEDD), left atrial dimension (LAD), left ventricular ejection fraction (LVEF), left ventricular posterior wall thickness (LVPWT), interventricular septum thickness (IVST), the peak gradient of the E-wave (E), the peak gradient of the A-wave (A). The E/A ratio was subsequently derived. Left ventricular volumes and left ventricular diastolic functions were not measured; instead, we focused on dimensions and ejection fraction as key parameters. Positive criteria for color Doppler ultrasound were defined as abnormal ultrasound cardiogram findings, including abnormal motion of the diseased wall relative to the opposite wall (*e.g.*, segmental hypokinesia or akinesia), absence of left ventricular wall segmental motion, localized hypokinesis, dyskinesis, and abnormal thickening during systole.

### Carotid artery ultrasound
Patients were positioned supine with the neck fully exposed. The probe frequency was adjusted to 7.5–10.0 MHz. The probe was moved along the direction of the common carotid artery, between the sternocleidomastoid muscle and the trachea, scanning continuously from top to bottom to display the proximal, distal, and mid-sections of the common carotid artery. Subsequently, the probe was gently shifted from the anterior aspect of the neck to the posterior region, traversing the carotid bifurcation in order to obtain images of the carotid arteries' long axis, showcasing both the external and internal carotid arteries. To ensure standardization, all ultrasonographers followed a standardized protocol for image acquisition and measurement, which included calibration checks before each session. Measurements were taken of the internal carotid artery's lumen diameter, while the echo intensity of the arterial wall was evaluated. Additionally, an assessment was conducted to identify any plaques, stenosis, or other morphological irregularities present within the lumen. For severely narrowed or occluded vessels, collateral circulation in the distal vessel was noted. The shape, size, distribution, and echo intensity of atherosclerotic plaques were
observed. The direction and velocity of blood flow, the presence of any flow defects or interruptions, and the location and degree of arterial stenosis were observed. Intima-media thickness (IMT) was recorded. Parameters related to blood flow were documented, encompassing diastolic velocity (Vd), systolic velocity (Vs), and the resistance index (RI). Positive criteria for carotid ultrasound were defined as follows: IMT < 1.0 mm as normal, 1.0 mm $\leq$ IMT < 1.5 mm as carotid atherosclerosis, and IMT $\geq$ 1.5 mm as local plaque formation. The Crouse score was determined by summing the maximum thicknesses of individual plaques detected in the carotid arteries (*Liu et al., 2025*).

## Follow up

Following their discharge, patients were monitored every two months through phone calls and routine outpatient visits. The follow-up period extended to three years to observe the incidence of major adverse cardiovascular events (MACE) during this timeframe. In this research, MACE was defined to include non-fatal myocardial infarction, unstable angina pectoris, arrhythmia, heart failure, stroke and all-cause mortality that occurred throughout the follow-up interval. Based on whether MACE events took place, participants were categorized into a non-MACE group ($n = 84$) and a MACE group ($n = 54$). Additionally, a separate validation cohort of 141 patients was established, which was also divided into a non-MACE group ($n = 90$) and a MACE group ($n = 51$). Patients in this validation cohort were selected based on the same inclusion and exclusion criteria as those in the primary cohort. All patients in the validation cohort also underwent the same standardized ultrasound procedures and follow-up protocols as described above. By employing a separate validation cohort, we aimed to ensure that our findings are robust and can be generalized to other patient populations. All patients included in this study cooperated with the follow-up, and their relevant data were complete. There was no missing data in this study.

## Statistical analysis

The statistical analysis was conducted utilizing SPSS version 26.0 (IBM Corp., Armonk, NY, USA). Continuous data that followed a normal distribution were reported as mean $\pm$ standard deviation and assessed between groups using the independent samples *t*-test. Categorical variables were represented as n (%) and analyzed for differences between groups *via* the chi-square ($\chi^2$) test. A *P*-value of less than 0.05 was deemed statistically significant. To evaluate the relationship between MACE and ultrasound parameters, Spearman correlation analysis was employed. Variables demonstrating significant differences in both differential and correlation assessments were factored in as covariates in the logistic regression analysis. Additionally, the area under the receiver operating characteristic (ROC) curve (AUC) was utilized to evaluate the predictive capability of ultrasound parameters concerning MACE.

Potential confounding factors such as age, gender, BMI, smoking history, alcohol consumption, type of chest pain, medication use, complication and lesion vessel numbers were identified *a priori*. To adjust for these confounders, we used multivariate logistic regression models. Each confounder was included in the model as a covariate, and

**Table 1  MACE situation.**

| MACE situation | n | Percentage |
|---|---|---|
| All-cause death | 18 | 13.04% |
| Non-fatal myocardial infarction | 14 | 10.14% |
| Unstable angina pectoris | 9 | 6.52% |
| Arrhythmia | 7 | 5.07% |
| Heart failure | 4 | 2.90% |
| Stroke | 2 | 1.45% |
| Total incidence rate | 54 | 39.13% |

their effects on the outcome were controlled for during the analysis. ROC curves were constructed to evaluate the predictive performance of our model. The optimal cut-off point for predicting MACE was determined using the Youden Index, which maximizes the sum of sensitivity and specificity. The cut-off point was chosen based on the highest Youden Index value, ensuring a balance between sensitivity and specificity.

# RESULTS

## MACE situation

This study ultimately included 138 coronary artery disease patients who met the inclusion criteria. The 3-year follow-up results after discharge showed that out of the 138 patients, 54 (39.13%) experienced MACE, including 18 all-cause deaths (13.04%), 14 non-fatal myocardial infarctions (10.14%), nine cases of unstable angina pectoris (6.52%), seven cases of arrhythmia (5.07%), four cases of heart failure (2.90%), and two cases of stroke (1.45%) (Table 1). Based on different prognoses during the follow-up period, patients were divided into a non-MACE group ($n = 84$, 60.87%) and a MACE group ($n = 54$, 39.13%).

## Comparison of basic information between the two groups

Table 2 presents the baseline characteristics of both patient groups. The findings indicate that there are no statistically significant differences in the baseline information between these groups ($P > 0.05$) (Table 2), suggesting that demographic and disease-related characteristics at baseline do not substantially influence the occurrence of MACE.

## Comparison of echocardiography related indicators between the two groups

An analysis of the echocardiography parameters between the two groups revealed that the MACE group exhibited substantially elevated LVEDD and notably reduced values for LVEF and A compared to the non-MACE group ($P < 0.05$) (Table 3). Conversely, no significant differences were observed in LVPWT, IVST, LVESD, LAD, $E$-value, or the E/A ratio between the two groups ($P > 0.05$).

## Comparison of carotid artery ultrasound related indicators between the two groups

When evaluating the carotid ultrasound parameters across the two groups, it was observed that the MACE group presented significantly elevated values for IMT and Crouse score

**Table 2  Comparison of basic information between the two groups.**

| Parameter | | non-MACE group ($n=84$) | MACE group ($n=54$) | $t/\chi^2$ | P value |
|---|---|---|---|---|---|
| Age (years) | | $67.13 \pm 10.69$ | $66.39 \pm 11.51$ | 0.381 | 0.704 |
| Gender (n (%)) | Male | 51 (60.71%) | 33 (61.11%) | 0.002 | 0.963 |
| | Female | 33 (39.29%) | 21 (38.89%) | | |
| BMI (kg/cm$^2$) | | $25.45 \pm 1.46$ | $25.71 \pm 1.23$ | 1.079 | 0.283 |
| Smoking history (n (%)) | | 24 (28.57%) | 18 (33.33%) | 0.352 | 0.553 |
| Drinking history (n (%)) | | 21 (25%) | 16 (29.63%) | 0.359 | 0.549 |
| Type of chest pain (n (%)) | AMI | 33 (39.29%) | 23 (42.59%) | 0.149 | 0.699 |
| | UA | 51 (60.71%) | 31 (57.41%) | | |
| Medications use (n (%)) | Aspirin | 65 (77.38%) | 40 (74.07%) | 0.198 | 0.657 |
| | Statins | 58 (69.05%) | 36 (66.67%) | 0.086 | 0.770 |
| Complication (n (%)) | Hypertension | 27 (32.14%) | 19 (35.19%) | 0.137 | 0.711 |
| | Diabetes | 8 (9.52%) | 8 (14.81%) | 0.898 | 0.343 |
| | Hyperlipidemia | 14 (16.67%) | 10 (18.52%) | 0.078 | 0.779 |
| Lesion vessel numbers (n (%)) | Single | 26 (30.95%) | 17 (31.48%) | 0.599 | 0.741 |
| | Double | 39 (46.43%) | 22 (40.74%) | | |
| | Three or more | 19 (22.62%) | 15 (27.78%) | | |

Notes.

BMI, Body mass index; AMI, acute myocardial infarction; UA, unstable angina.

**Table 3  Comparison of echocardiography related indicators between the two groups.**

| Item | non-MACE group ($n=84$) | MACE group ($n=54$) | $t$ | P value |
|---|---|---|---|---|
| LVPWT (mm) | $10.24 \pm 3.83$ | $11.27 \pm 3.61$ | 1.575 | 0.117 |
| IVST (mm) | $10.25 \pm 3.36$ | $11.22 \pm 3.64$ | 1.613 | 0.109 |
| LVESD (mm) | $22.53 \pm 4.48$ | $23.26 \pm 5.33$ | 0.859 | 0.392 |
| LVEDD (mm) | $53.68 \pm 6.52$ | $56.82 \pm 8.39$ | 2.335 | 0.022 |
| LAD (mm) | $40.15 \pm 5.31$ | $41.91 \pm 5.06$ | 1.931 | 0.056 |
| LVEF (%) | $53.24 \pm 10.74$ | $48.72 \pm 9.41$ | 2.526 | 0.013 |
| E-value (cm/s) | $78.13 \pm 10.27$ | $77.65 \pm 11.67$ | 0.255 | 0.799 |
| A-value (cm/s) | $55.45 \pm 8.56$ | $52.26 \pm 8.12$ | 2.177 | 0.031 |
| E/A | $1.28 \pm 0.34$ | $1.34 \pm 0.29$ | 1.062 | 0.290 |

Notes.

LVPWT, Left ventricular posterior wall thickness; IVST, interventricular septum thicknes; LVESD, left ventricular end-systolic diameter; LVEDD, left ventricular end-diastolic dimension; LAD, left atrial dimension; LVEF, left ventricular ejection fraction; E-value, peak gradient-E wave; A-value, peak gradient-A wave; E/A, peak gradient-E wave/peak gradient-A wave.

**Table 4 Comparison of carotid artery ultrasound related indicators between the two groups.**

| Item | Non-MACE group (n = 84) | MACE group (n = 54) | t | P value |
|---|---|---|---|---|
| IMT (mm) | 1.67 ± 0.21 | 1.76 ± 0.17 | 2.693 | 0.008 |
| Vs (cm/s) | 39.25 ± 7.78 | 42.34 ± 10.31 | 1.885 | 0.063 |
| Vd (cm/s) | 16.29 ± 3.46 | 17.37 ± 5.47 | 1.303 | 0.196 |
| RI | 0.65 ± 0.11 | 0.68 ± 0.09 | 1.588 | 0.115 |
| Crouse integral | 0.75 ± 0.07 | 0.79 ± 0.11 | 2.369 | 0.020 |

Notes.
IMT, intima-media thickness; Vs, velocity systolic; Vd, velocity diastolic; RI, resistance index.

**Table 5 Multivariate logistic analysis of MACE occurrence in elderly patients with coronary heart disease.**

| Independent variable | SE | Wald $\chi^2$ | OR value | 95% CI | P value |
|---|---|---|---|---|---|
| LVEDD (mm) | 0.028 | 2.389 | 1.069 | 1.012–1.129 | 0.017 |
| LVEF (%) | 0.019 | −1.977 | 0.964 | 0.929–1.000 | 0.048 |
| A-value (cm/s) | 0.023 | −2.155 | 0.951 | 0.909–0.995 | 0.031 |
| IMT (mm) | 1.049 | 2.420 | 12.643 | 1.619–98.718 | 0.016 |
| Crouse integral | 2.235 | 2.261 | 156.405 | 1.958–12,490.786 | 0.024 |

Notes.
LVEDD, left ventricular end-diastolic dimension; LVEF, left ventricular ejection fraction; A-value, peak gradient-A wave; IMT, intima-media thickness.

in comparison to the non-MACE group ($P < 0.05$) (Table 4). In contrast, no significant differences were detected in Vs, Vd, or RI between the two groups ($P > 0.05$).

## Multivariate logistic analysis of MACE occurrence in elderly patients with coronary artery disease

Using the occurrence of MACE as the dependent variable (coded as occurrence = 1, non-occurrence = 0), a multivariate logistic regression analysis was conducted with the echocardiography parameters (LVEDD, LVEF, and A value) and carotid ultrasound parameters (IMT and Crouse score, all considered as actual measurement values) as independent variables. The analysis revealed that LVEDD, IMT, and Crouse score were identified as risk factors for MACE in elderly individuals with coronary artery disease (OR > 1), whereas LVEF and A value were found to be protective factors against the occurrence of MACE (OR < 1) (Table 5).

## Predictive value of combining echocardiography with carotid ultrasound for MACE in elderly patients with coronary artery disease

To evaluate the predictive performance of various parameters for MACE in elderly patients with coronary artery disease, we conducted ROC analysis. The optimal cut-off points were determined using the Youden index, which maximizes the sum of sensitivity and specificity (Table 6). In this study, the optimal thresholds of various parameters and their Youden indices were as follows: LVEDD (62.025 mm, Youden index 0.269), LVEF (51.635%, Youden index 0.316), A value (49.8 cm/s, Youden index 0.249), IMT (1.585 mm, Youden

**Table 6  ROC analysis of MACE occurrence in elderly patients with coronary heart disease.**

|  | Best_threshold | Sensitivities | Specificities | AUC | Youden_index | F1_score |
|---|---|---|---|---|---|---|
| LVEDD (mm) | 62.025 | 0.352 | 0.917 | 0.634 | 0.269 | 0.475 |
| LVEF (%) | 51.635 | 0.685 | 0.631 | 0.627 | 0.316 | 0.274 |
| A-value (cm/s) | 49.76 | 0.463 | 0.786 | 0.605 | 0.249 | 0.389 |
| IMT (mm) | 1.585 | 0.907 | 0.369 | 0.625 | 0.276 | 0.628 |
| Crouse integral | 0.825 | 0.352 | 0.869 | 0.603 | 0.221 | 0.452 |

**Notes.**

LVEDD, left ventricular end-diastolic dimension; LVEF, left ventricular ejection fraction; A-value, peak gradient-A wave; IMT, intima-media thickness.

index 0.276), and Crouse score (0.825, Youden index 0.221). Although these parameters showed some predictive value, they did not perform notably when used individually. Future research should consider combining multiple parameters or developing composite scores to improve the accuracy of MACE prediction.

The area under the ROC curve for the combined prediction of MACE using LVEDD, LVEF, A-value, IMT, and Crouse score was 0.755 (Fig. 1), indicating good predictive value for MACE in elderly patients with coronary artery disease when combining echocardiography with carotid ultrasound.

### General information (validation set)

In the validation set, 90 patients did not experience MACE, while 51 patients did. There were no significant differences in baseline information between the two groups ($P > 0.05$) (Table 7), suggesting that other factors may have influenced the occurrence of MACE.

### Predictive indicators (validation set)

In the validation set, echocardiography parameters and carotid ultrasound parameters also showed significant differences. Among them, the MACE group had significantly higher LVEDD, IMT, and Crouse score and significantly lower LVEF and A-value compared to the non-MACE group ($P < 0.05$) (Table 8). This indicates that these indicators have a significant impact on the occurrence of MACE.

### Joint model (validation set)

A combined predictive model for MACE was constructed using echocardiography parameters and carotid ultrasound parameters. The results showed that the AUC of the combined model in the validation set was 0.754 (Fig. 2), indicating good predictive value for the occurrence of MACE using the combination of echocardiography and carotid ultrasound parameters.

### DISCUSSION

Coronary artery disease progresses over time, affecting patients throughout their lifetime and is a leading cause of death globally (*Hasbani et al., 2022*; *Katta et al., 2021*; *Manohar et al., 2022*). The prognosis of CHD patients is closely related to the number of diseased coronary arteries and the severity of lesions. Early diagnosis and intervention are critical for improving prognosis, but currently, there is a lack of specific markers for assessing the

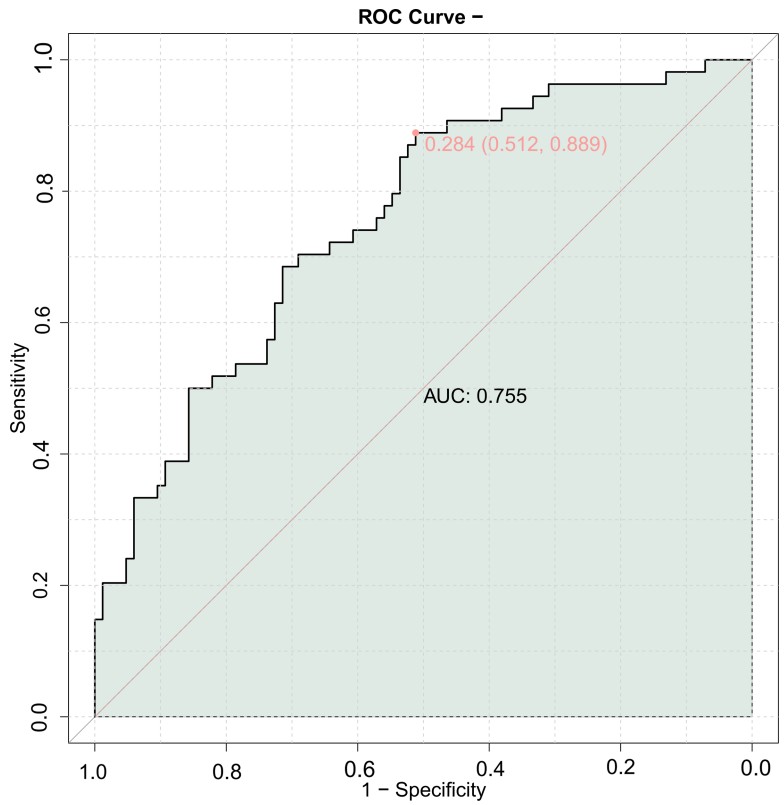

**Figure 1** **Receiver operating characteristic (ROC) curve of combining echocardiography with carotid ultrasound for major adverse cardiovascular events (MACE) in elderly patients with coronary heart disease.**

prognosis of CHD. This study aimed to evaluate the predictive accuracy of a combined model using cardiac and carotid ultrasound parameters for MACE in elderly CHD patients. Our results indicate that the combination of LVEDD, LVEF, A value, IMT, and Crouse score has good predictive value for MACE in this patient population.

The echocardiographic findings from this study indicated that the LVEDD was markedly greater in the MACE group than in the non-MACE group, whereas both LVEF and A values were significantly reduced in the MACE group. Moreover, LVEDD is a risk factor for MACE in elderly CHD patients, whereas LVEF and A values are protective factors against MACE. This is consistent with previous studies, underscoring the importance of echocardiography in assessing MACE (*Zhang et al., 2021*). For example, *Peng et al. (2022)* reported have shown that decreased LVEF and increased LVEDD are related with an increased risk of MACE. LVEDD and LVEF are important indicators of left ventricular systolic and diastolic function in CHD patients (*Han et al., 2020*; *Gao et al., 2020*; *Pezawas et al., 2017*). The A value can well reflect blood flow velocity and microvascular density, providing quantitative assessment of myocardial blood flow, which also reflects left ventricular diastolic function (*Mishra et al., 2011*). This indicates that LVEDD, LVEF, and A value have significant clinical

**Table 7  Comparison of basic information between the two groups.**

| Parameter | | Non-MACE group (n = 90) | MACE group (n = 51) | t/χ² | P value |
|---|---|---|---|---|---|
| Age (years) | | 66.54 ± 6.78 | 67.25 ± 7.14 | 0.583 | 0.561 |
| Gender (n (%)) | Male | 55 (61.11%) | 38 (74.51%) | 2.603 | 0.107 |
| | Female | 35 (38.89%) | 13 (25.49%) | | |
| BMI (kg/cm²) | | 24.26 ± 2.15 | 24.35 ± 2.34 | 0.232 | 0.817 |
| Smoking history (n (%)) | | 25 (27.78%) | 17 (33.33%) | 0.480 | 0.488 |
| Drinking history (n (%)) | | 25 (27.78%) | 16 (31.37%) | 0.204 | 0.652 |
| Type of chest pain (n (%)) | AMI | 31 (34.44%) | 22 (43.14%) | 1.049 | 0.306 |
| | UA | 59 (65.56%) | 29 (56.86%) | | |
| Medications use (n (%)) | Aspirin | 66 (73.33%) | 39 (76.47%) | 0.169 | 0.681 |
| | Statins | 59 (65.56%) | 35 (68.63%) | 0.138 | 0.710 |
| Complication (n (%)) | Hypertension | 30 (33.33%) | 14 (27.45%) | 0.525 | 0.469 |
| | Diabetes | 8 (8.89%) | 6 (11.76%) | 0.301 | 0.583 |
| | Hyperlipidemia | 14 (15.56%) | 7 (13.73%) | 0.086 | 0.769 |
| Lesion vessel numbers (n (%)) | Single | 27 (30.00%) | 17 (33.33%) | 0.196 | 0.907 |
| | Double | 40 (44.44%) | 21 (41.18%) | | |
| | Three or more | 23 (25.56%) | 13 (25.49%) | | |

Notes.
   BMI, Body mass index; AMI, acute myocardial infarction; UA, unstable angina.

**Table 8  Predictive indicators (validation set).**

| Parameter | Non-MACE group (n = 90) | MACE group (n = 51) | t | P value |
|---|---|---|---|---|
| LVEDD (mm) | 52.47 ± 6.28 | 55.16 ± 6.37 | 2.434 | 0.016 |
| LVEF (%) | 51.59 ± 8.15 | 48.24 ± 8.36 | 2.322 | 0.022 |
| A-value (cm/s) | 53.16 ± 7.45 | 49.47 ± 8.24 | 2.719 | 0.007 |
| IMT (mm) | 1.66 ± 0.34 | 1.79 ± 0.31 | 2.219 | 0.028 |
| Crouse integral | 0.75 ± 0.14 | 0.85 ± 0.23 | 2.774 | 0.007 |

Notes.
   LVEDD, left ventricular end-diastolic dimension; LVEF, left ventricular ejection fraction; A-value, peak gradient-A wave; IMT, intima-media thickness.

value in prognostic assessment, consistent with the findings reported in the literature (*Gong & Li, 2016*; *Zhang & Shi, 2023*).

The pathological process leading to MACE in CHD patients is complex, with heart dysfunction being just one of the important causes; therefore, prediction based solely on a single echocardiographic parameter has certain limitations. The latest ESC guidelines emphasize the usefulness of carotid ultrasound in diagnosing coronary artery disease and subclinical atherosclerosis assessment, which helps determine cardiovascular risk levels (*Artyszuk et al., 2023*). Our study also supports the use of carotid ultrasound as a supplementary tool for risk stratification in CHD patients. The results from carotid ultrasound assessments in this study revealed that the IMT and Crouse score were significantly elevated in the MACE group compared to those in the non-MACE group.

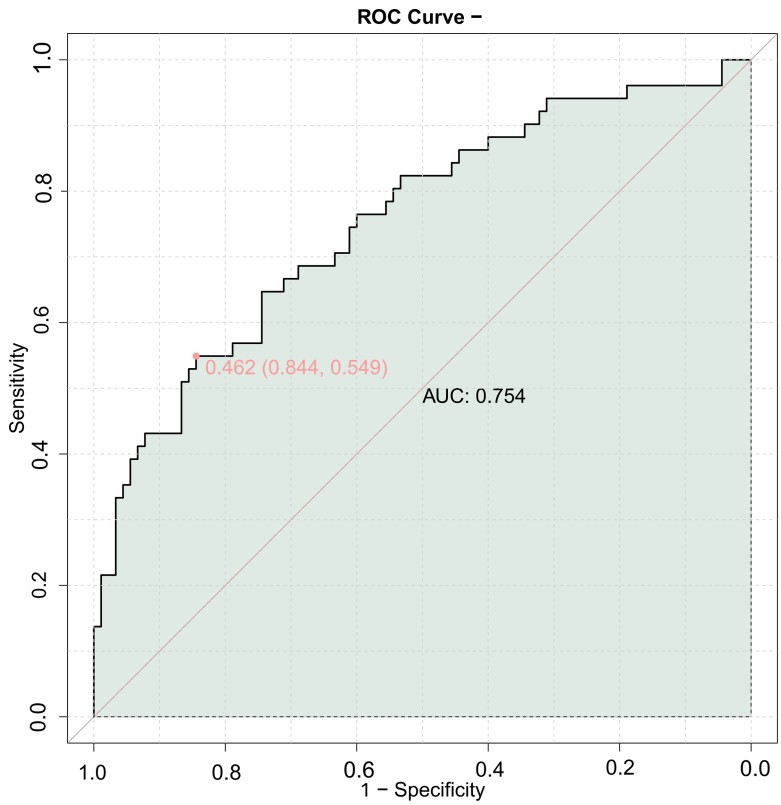

**Figure 2** ROC curve of combining echocardiography with carotid ultrasound for MACE in elderly patients with coronary heart disease (validation set).

Furthermore, both IMT and the Crouse score were identified as risk factors for MACE in elderly patients with CHD. These results align with findings from earlier studies, highlighting the potential of carotid ultrasound as a predictor for cardiovascular events (*Bao et al., 2023*). For example, *Ling et al. (2023)* report indicated that increased IMT is a predictive factor for future cardiovascular events. Carotid ultrasound can measure the patient's carotid IMT, blood flow velocity, and determine the presence of plaque formation, thereby indirectly reflecting coronary artery hardening (*Qi, Cheng & Zhang, 2004*). Higher IMT suggests the presence of carotid stenosis or occlusion, which may lead to slowed blood flow or embolus formation. Increased IMT is a precursor and pathological basis for plaque formation, which in later stages can develop into atherosclerotic plaques. As the disease progresses, it can gradually evolve into organized thrombi, intraplaque hemorrhage, lipid deposition, fibrosis, or calcification (*Song et al., 2021*). The carotid Crouse score is an important indicator for predicting cerebrovascular accidents, and as the condition worsens, the carotid Crouse score in CHD patients significantly increases, possibly related to further exacerbation of carotid stenosis in CHD patients. Clinicians should closely monitor changes in the condition to prevent the occurrence of MACE (*Zaid et al., 2017*).

The combined predictive model using cardiac and carotid ultrasound parameters showed an AUC of 0.755 for predicting MACE with the LVEDD, LVEF, A value, IMT, and Crouse

score, and a validation set AUC of 0.754. These similar results indicate that the combined detection of these five indicators has good predictive value for the prognosis of elderly CHD patients, allowing for more effective guidance of clinical work and identification of patients with poor prognosis.

While this study offers valuable new perspectives by integrating ultrasound parameters in the evaluation of major adverse cardiovascular events among older individuals with coronary artery disease, it is crucial to acknowledge certain limitations. The primary limitation stems from the study being a single-center retrospective design with a limited sample size, which may introduce selection bias and affect the generalizability of our findings. The follow-up period was restricted to 3 years, which precludes an assessment of long-term outcomes. To enhance the robustness of our findings, future studies should aim to increase the sample size, address potential biases through more rigorous methodologies, and conduct prospective clinical trials to validate the results.

Furthermore, we did not perform multicollinearity diagnostics such as Variance Inflation Factor (VIF) in this study. This omission could potentially affect the reliability of our regression models, as high multicollinearity can lead to unstable estimates of regression coefficients. Future studies should consider assessing multicollinearity to ensure the robustness of predictive models. Despite this limitation, our results provide valuable insights into predicting MACE among older individuals with coronary artery disease.

## CONCLUSION

In summary, this study suggests that a combination of cardiac and carotid ultrasound parameters can predict MACE within 3 years in elderly CHD patients. Through cardiac and carotid ultrasound examination, not only can the diagnostic rate of cardiovascular diseases be improved, but the prognosis of CHD patients can also be predicted, providing objective evidence for clinicians to identify high-risk CHD patients early.

### Funding
The authors received no funding for this work.

### Competing Interests
The authors declare there are no competing interests.

### Author Contributions

- Tianli Jia conceived and designed the experiments, performed the experiments, prepared figures and/or tables, authored or reviewed drafts of the article, and approved the final draft.
- Xiaoxiao Li conceived and designed the experiments, performed the experiments, prepared figures and/or tables, and approved the final draft.
- Qianqian Xie analyzed the data, authored or reviewed drafts of the article, and approved the final draft.

## Human Ethics

The following information was supplied relating to ethical approvals (i.e., approving body and any reference numbers):

This study was approved by Yantai Yuhuangding Hospital's ethics committee.

## Data Availability

The data is available in the Supplementary Files.

## Supplemental Information

Supplemental information for this article can be found online at http://dx.doi.org/10.7717/peerj.19688#supplemental-information.

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
