# Peer review of "Combining echocardiography with carotid ultrasound parameters in predicting major adverse cardiovascular events among older individuals with coronary artery disease"

_PeerJ, doi:10.7717/peerj.19688_

## Round 0.1 · original submission · Major Revisions

The manuscript addresses an important topic of predicting major adverse cardiovascular events in elderly coronary artery disease patients using a combined model of cardiac and carotid ultrasound parameters. While the study is well-conceived, several areas require attention:

Language and Typographical Errors: Numerous typographical issues (e.g., garbled characters in tables) and grammatical inconsistencies should be corrected for clarity and professionalism.

Statistical Analysis: Clarify how multicollinearity was assessed and whether adjustments for confounders were made. Justify specific cut-off points in ROC analysis.

Figures and Tables: Improve clarity in ROC curve labels and ensure consistent formatting in tables.

Discussion and Limitations: Better highlight the novelty of combining ultrasound parameters and address selection bias, missing data, and short follow-up duration in the limitations.

Reproducibility: Provide details on standardization of ultrasound measurements and selection criteria for the validation cohort.

These revisions will strengthen the manuscript’s clarity, rigor, and impact.

Reviewer 1 ·

Basic reporting

A very interesting manuscript incorporating two important techniques in the management of athersclerosis related complications.
In the introduction you state that the mainstay of coronary artery disease management is PCI which is true in acute coronary sydromes but is medically managed in chronic coronary syndromes.
I would recommend using the terminology of echocardiography not cardiac ultrasound.

Experimental design

You could have said that the patients included were those with an acute coronary syndrome. The inclusion criteria are not clearly defined.
Did you exclude those with a previous PCI or CABG?
Did you exclude those with heart failure?
Did you exclude those with a previous stroke?
Did you measure left ventricular volumes or just dimensions?
Did you assess the left ventricular diastolic functions?

Validity of the findings

Were there no incidences of stroke?

Reviewer 2 ·

Basic reporting

No comment

Experimental design

1. This study retrospectively analyzed coronary heart disease patients hospitalized between June 2019 and June 2020. This means the method of this study is cohort retrospective. However, the author divided the group into MACE and non-MACE and then identified the determinant factors based on those two groups, which seems more like a case-control study. This study design will affect the recruitment bias, selection bias, and reporting bias of the study.
2. The authors also excluded lost-to-follow-up patients (lines 65-66). Information on the number of lost-to-follow-up patients is needed to identify the quality of the study. If the authors did a cohort retrospective, then it is better to use a flow diagram to show the number of recruited patients, the number of loss-to-follow-up patients, and the number of final participants.
3. What is the meaning of separate validation cohort number of patients?

Validity of the findings

1. The basic information offered in this study is very limited. Since this is a long observation study, so we need to make sure the baseline characteristics of patients are in equal condition.
2. The authors also did not define the type of chest pain in the included participants. There is no information on how many of the patients included in this study started with the condition of myocardial infarction or with unstable angina since it will have a different prognosis. The authors also did not mention the acute management of these patients when the patients got chest pain in the early of the study, whether the patients already had full revascularization or not, or even needed to get bypass surgery.
2. There is no information about the medicines that are routinely consumed by the patients that can affect the prognosis of the patients.

Additional comments

1. Line 69-70 “As the study did not involve human intervention, the ethics committee waived the patient's informed consent.” Even to use patient data for research, we need to have informed consent with the patients.
2. Patients included in this study are those who had chest pain and were diagnosed with either acute myocardial infarction of unstable angina. However, one of the MACE defined in this study is angina pectoris. Is it stable angina pectoris or unstable angina pectoris?, please define it clearly.

---

## Round 0.2 · Minor Revisions

Dear Author,

Thanks a lot for the meticulous revision of your manuscript. However, there are a few more important things that need your attention before I recommend the article for publication.

First: The referencing of the manuscript does not match what is recommended by the PeerJ journal. [STAFF NOTE: This will be addressed in the typesetting stage if the article is accepted]

Second: Correct these grammatical errors.

3. Running title should reflect study aim → change to “Combined ultrasound prediction of MACE.”

6 “Electroencephalogram Room” is unrelated to cardiology → replace with correct cardiology/cardiac-ultrasound department name.

9–10 Re-phrase: “Tianli Jia and Xiaoxiao Li contributed equally to this work.”

16 Replace “Purpose:” with “Objective:” and delete second “parameters” (“…combining echocardiography with carotid ultrasound in predicting…”).

19–58 (multiple) Use one disease term consistently: choose either “coronary artery disease (CAD)” or “coronary heart disease (CHD).”

20–22 vs. 151–152 Ensure sample counts are identical (94 + 44 vs. 84 + 54).

27–30 Combine significance wording once: “The MACE group had higher LVEDD, IMT, and Crouse score, and lower LVEF and A-wave velocity (all P < 0.05).”

31–32 Re-state AUCs succinctly: “Derivation AUC = 0.755; validation AUC = 0.754.”

40 Correct typo: “he management” → “The management.”

50 Make causal claim cautious: “may reflect the degree of myocardial ischemia.”

56–58 Hedge absolute claim: change to “Few studies have examined combined use for MACE prediction.”

64 Design label: replace “case-control” with “retrospective cohort.”

68 Clarify: “Patients diagnosed with acute myocardial infarction (AMI) or unstable angina (UA).”

70 Grammar: “(5 case)” → “(5 cases).”

80 Explain or remove timing detail (“on hospital day 2”) unless justified clinically.

89–90 Fix clause: “…A-wave (A). The E/A ratio was subsequently derived.”

92–94 Specify abnormal wall motion definition more precisely (e.g., “segmental hypokinesia or akinesia”).

108–113 Standardize inequalities: “IMT ≤ 1.0 mm; 1.0 < IMT ≤ 1.2 mm; … IMT ≥ 4.1 mm.”

115–120 Align group sizes here with earlier counts to avoid contradiction.

128 SPSS citation once: “SPSS v26.0 (IBM Corp., Armonk, NY, USA).”

137–139 Wording: “medication use” not “medications use”; “alcohol consumption” not “drinking history.”

166–167 Define Vs, Vd, RI at first appearance.

181 Round A-wave cut-off to one decimal (e.g., 49.8 cm/s).

186 Choose one term consistently: “Crouse score” (or “integral”), not both.

215 Verb tense: “Peng et al. reported.”

242–244 Split run-on: “…AUC of 0.754. These similar results indicate…”

253–256 State clearly whether multicollinearity diagnostics (e.g., VIF) were performed; keep message consistent with earlier methods.

Regards

Reviewer 1 ·

Basic reporting

I thank the authors for their response to all queries raised. Their response was quick and adressed all points raised. There should be a final grammar and language check.

Experimental design

The flaws in the design were answered.

Validity of the findings

The validity of findings is fine which was expected by the nature of cases and the investigations used. So, really nothing innovative.

---

## Round 0.3 · accepted · Accept

I am satisfied with the amendments made by the author, and the paper is ready for publication. However, minor grammatical errors can be corrected during publication process.